# Classification of Postprandial Glycemic Status with Application to Insulin Dosing in Type 1 Diabetes—An In Silico Proof-of-Concept

**DOI:** 10.3390/s19143168

**Published:** 2019-07-18

**Authors:** Giacomo Cappon, Andrea Facchinetti, Giovanni Sparacino, Pantelis Georgiou, Pau Herrero

**Affiliations:** 1Department of Information Engineering, University of Padova, 35131 Padova (PD), Italy; 2Department of Electrical and Electronical Engineering, Imperial College London, London W5 5SA, UK

**Keywords:** continuous glucose monitoring, decision support systems, machine learning, type 1 diabetes, gradient boosted trees, postprandial glycaemia

## Abstract

In the daily management of type 1 diabetes (T1D), determining the correct insulin dose to be injected at meal-time is fundamental to achieve optimal glycemic control. Wearable sensors, such as continuous glucose monitoring (CGM) devices, are instrumental to achieve this purpose. In this paper, we show how CGM data, together with commonly recorded inputs (carbohydrate intake and bolus insulin), can be used to develop an algorithm that allows classifying, at meal-time, the post-prandial glycemic status (i.e., blood glucose concentration being too low, too high, or within target range). Such an outcome can then be used to improve the efficacy of insulin therapy by reducing or increasing the corresponding meal bolus dose. A state-of-the-art T1D simulation environment, including intraday variability and a behavioral model, was used to generate a rich in silico dataset corresponding to 100 subjects over a two-month scenario. Then, an extreme gradient-boosted tree (XGB) algorithm was employed to classify the post-prandial glycemic status. Finally, we demonstrate how the XGB algorithm outcome can be exploited to improve glycemic control in T1D through real-time adjustment of the meal insulin bolus. The proposed XGB algorithm obtained good accuracy at classifying post-prandial glycemic status (AUROC = 0.84 [0.78, 0.87]). Consequently, when used to adjust, in real-time, meal insulin boluses obtained with a bolus calculator, the proposed approach improves glycemic control when compared to the baseline bolus calculator. In particular, percentage time in target [70, 180] mg/dL was improved from 61.98 (±13.89) to 67.00 (±11.54; *p* < 0.01) without increasing hypoglycemia.

## 1. Introduction

Type 1 diabetes (T1D) is a lifelong condition characterized by the destruction of the pancreatic beta cells responsible for insulin production. Consequently, people affected by T1D strongly rely on exogenous insulin in order to maintain blood glucose (BG) levels within the safe range ([70, 180] mg/dL) [1] and avoid long-term complications, e.g., neuropathy, retinopathy, and micro/macro-vascular heart diseases, due to sustained hyperglycemia (BG > 180 mg/dL) [2]. However, administration of exogenous insulin is difficult to tune and an excess of it can cause hypoglycemia (BG < 70 mg/dL), which leads to dizziness, lightheadedness, fainting, and, in the most extreme cases, coma or even death.

Insulin is exogenously injected either via multiple daily injections (MDI), consisting of fast-acting insulin performed at meal-time (bolus insulin) and a long-acting dose once (or twice) per day, or via an infusion pump (i.e., continuous subcutaneous insulin infusion). Titration of such doses is usually done according to self-monitoring of blood glucose (SMBG), diet, and physical activity, among others [1]. In particular, the calculation of the meal-time insulin bolus is usually done by applying the standard bolus calculator formula, which is commonly available in most insulin pumps and some glucose meters. However, due to unaccounted physiological events and variability, this formula is, in many cases, suboptimal and might lead to post-prandial hyper- and hypoglycemia [3]. In the past few years, exciting perspectives in the management of T1D were opened by the introduction of new technologies, such as minimally invasive sensors for real-time continuous glucose monitoring (CGM). CGM has revolutionized BG monitoring providing a measurement every 1–5 min, mitigating the need of the SMBG and greatly increasing the information on BG fluctuations [4,5]. CGM and additional data streams obtained from wearable devices, such as activity monitors, can provide important real-time information on the current status of the diabetic individual, and appear to be suitable to develop new “smarter” tools to empower people with T1D with decision support to improve the management of their condition [6,7]. In recent years, several methodologies have been proposed with the aim of optimizing the insulin bolus dose by using the subject’s information at meal-time (e.g., BG level, carbohydrate intake, previous insulin bolus, activity counts, and therapy parameters), and have been demonstrated to improve glycemic control [8,9]. Furthermore, additional improvements can be achieved if, in addition to using information at meal-time, a prediction of postprandial glycemic status, i.e., future BG being too low, too high, or within a target range; is accounted for [10,11].

In the present work, we develop a classification technique usable at meal-time to predict the future glycemic status in the postprandial period (i.e., between 2 h and 6 h after meal) exploiting the knowledge of pre-prandial CGM measurements, carbohydrate intake estimates and insulin infusion recordings. In particular, the technique uses the extreme gradient-boosted tree (XGB) algorithm [12] to classify whether the postprandial glycemic status is expected to be too high, too low, or within a target range. Finally, to show the potential usefulness of the proposed classification methodology in the daily management of T1D insulin therapy, we demonstrate that, by using a simple heuristic technique, an improvement on glycemic control can be obtained by adjusting the insulin bolus according to the predicted glycemic status.

Assessment of the method is performed by resorting to clinical trials. This is a strategy widely employed by the diabetes technology scientific community to develop and evaluate new algorithms before moving on to real subjects [13,14,15]. In particular, here we use synthetic data generated with a state-of-the-art mathematical model of a T1D subject decision-making (T1D-DM) [16]. This model is an evolution of a software tool that received approval by the US Food and Drug Administration (FDA) as a substitute for pre-clinical animal testing [17], which in addition to the glucose-insulin-glucagon dynamics, includes intraday variability of insulin sensitivity and a behavioral model for a T1D population. 

## 2. Method to Classify Future Postprandial Glycemic Status

### 2.1. Classification Criterion

To define the glycemic status in the post-prandial time window, i.e., if after the meal the individual glycemia will be in hypoglycemia, hyperglycemia, or within a target range, the CGM-based metric described by Herrero et al. [18] was employed. It consists in computing the minimum postprandial CGM (*G_min_*) level within a predefined postprandial time window as:(1)$$G_{min}\;=\;\underset{t\in \left[t_{m}+\;2h,\;t_{m}+\;6h\right]}{\min}CGM\left(t\right),$$
where, *t_m_* is the time of the meal. The rational for choosing a postprandial time window from two to six hours after meal time is that we are interested in computing the minimum postprandial glucose after the glucose peak time. Note that a typical glucose peak after meals is around 70 min but can potentially be longer for low-absorption meals. We then chose to end the window at six hours post meal to make sure that the effect of the short-acting meal insulin bolus is over [19]. Finally, the assumption that basal insulin (i.e., long-acting insulin) is correctly adjusted is made.

Then, the classification target *Y* consists of three classes defined as follows:(2)$$Y=\{\begin{array}{c} C_{1},\;G_{min}\;<\;{\mathrm{th}}_{\mathrm{hypo}},\\ C_{2},\;{\mathrm{th}}_{\mathrm{hypo}}\;\leq \;G_{min}\;\leq \;{\mathrm{th}}_{\mathrm{hyper}}\\ C_{3},\;G_{min}\;>\;{\mathrm{th}}_{\mathrm{hyper}}, \end{array},$$
where th_hypo_ and th_hyper_ are the hypoglycemia and hyperglycemia thresholds, respectively. Here, these thresholds were set to th_hypo_ = 70 mg/dL and th_hyper_ = 140 mg/dL. Figure 1 shows a graphical representation of the employed classification criterion.

In this context, an extreme gradient-boosted tree model, hereafter referred to as XGB, was chosen to classify the postprandial excursion into classes C_1_, C_2_, or C_3_.

### 2.2. Extreme Gradient-Boosted Tree Model

XGB is a highly effective and efficient supervised learning algorithm that has been shown to provide state-of-the-art results in many classification tasks [12]. It is particularly well known for being robust against outliers, to work well for relatively small datasets, and to automatically handle feature selection; thus, it well fitted the purpose of this work. Moreover, XGB associates each input to the probability of belonging to each one of the three considered target classes, i.e., p(C_1_), p(C_2_), and p(C_3_). This feature will result in being very useful, as described later in Section 3.

For each studied subject *p*, we trained a different XGB model, XGB*_p_*, the aim being accounting for inter-subject variability and personalizing the methodology at the individual level. For this purpose, a software framework (schematized in Figure 2) was built to automatically manage data preparation, model tuning, and testing.

#### 2.2.1. Features Vector and Data Preparation

The selected input (features vector; *X_ip_*) for the XGB model is composed by the following entities:Estimated amount of ingested carbohydrates (CHO*_i_*);Meal insulin bolus (IMB*_i_*);Two binary indicators denoting whether there was a hypo/hyperglycemic event in the last three hours. This feature allows us to capture the physiological response to hypoglycemia (e.g., secretion of glucagon) or the ingestion of rescue carbohydrates;The hour-of-day of *t_mi_* and three binary indicators representing the meal type (i.e., breakfast, lunch, or dinner), which are used to capture the subject’s intra-day variability (e.g., circadian rhythms);Two features describing the time elapsed since the last insulin bolus and meal intake, respectively. This feature might help to capture specific patient behaviors, such as using multiple boluses to treat the same meal and/or snacking pattern;CGM data within the time window (*t_m_**_i_* – 1 h, *t_m_**_i_*); in addition, data were preprocessed in order to obtain additional features. In detail, for each ingested meal at time *t_mi_*, CGM data, the estimated amount of carbohydrates (CHO), and insulin data (INS), were considered within the time window (*t_mi_* – 1 h, *t_mi_*). Then, such data were processed as follows:CGM was used to obtain the corresponding glucose rate of change, static risk (SR), and dynamic risk (DR) [20] time series, which empower the model with additional features that capture the dynamics of the CGM signal (e.g., glycemic variability);CHO was used to calculate the rate of glucose appearance in the blood (Ra) within (*t_mi_, t_mi_* + 1 h) through the use of a gastrointestinal model [21] to describe carbohydrate digestion and glucose absorption;INS data were transformed into two continuous signals representing an estimate of plasma insulin concentration (IP) [22] and the insulin-on-board (IOB) [23] to account for insulin absorption and clearance. As per the Ra signal, IP and IOB were estimated within (*t_m_**_i_*, *t_m_**_i_* + 1 h) assuming no additional insulin infusion was in that period.

#### 2.2.2. Model Tuning and Testing

For each subject, training of XGB*_p_* was performed through the gradient descent algorithm [24]. Figure 2 schematizes the model tuning and testing procedure. In detail, referring to Figure 2, Block A splits the dataset (*X_ip_*, *Y_ip_*) into training and testing data. Block A was also in charge of initializing the hyper-parameters to random values. These hyper-parameters are: The number of trees, the maximum depth of each tree, the subsample ratio of the training instances, the L2 regularization term on weights, and the learning rate. Then, in Block B, training data were used to tune the model and, given a set of hyper-parameters *h*, performance was assessed in a three-fold cross validation setting. Specifically, training data were split in three folds, of which two folds were used for training, and the third one was used to validate and evaluate the model. Then, performance of the *k*-th hyper-parameter set was quantified in terms of intra-folds macro-average area under the receiver operator characteristic curve (AUROC) [25]:(3)$$CVAUROC_{k}\;=\;\frac{1}{3}\sum_{j\;=\;1}^{3}AUROC_{macro_{j}},$$
where *AUROC_macroj_* denotes the intra-folds macro-average AUROC computed using the *j*-th fold as the validation set:(4)$$AUROC_{macro_{j}}\;=\;\sum_{c\;=\;1}^{3}AUROC_{C_{c}},$$
where *AUROC_Cc_* stands for the AUROC of the *c*-th class computed in a one-vs.-all fashion. 

Hyper-parameters were optimized by iterating Block C. Specifically, it implemented the tree-structured Parzen estimator (TPE) technique [26], i.e., an optimization algorithm that, for each iteration, collects a new observation (the *k*-th set of hyper-parameters and the respective performance *CVAUROC_k_*) and selects which set of hyper-parameters should be tried in the next iteration in order to improve the model performance. Finally, Block D selected the set of hyper-parameters *k* associated to the maximum *CVAUROC_k_* and it obtained the predicted classes on the test set.

### 2.3. Simulated Dataset

To evaluate the proposed XGB model, data of 100 virtual adult subjects were generated by simulation over two months using T1D-DM, which expands the model implemented in the University of Virginia/Padova T1D Simulator [17] with new modules describing glucose sensors error [27,28], intraday variability of insulin sensitivity [29] and patient behavior. Of note, physical activity was not included in the simulations, since the current version of the University of Virginia/Padova T1D simulator does not include a mathematical model describing its impact on glucose metabolism. More specifically, it consists of a large-scale maximal mathematical computer model of glucose, insulin, and glucagon dynamic of people with T1D, able to describe the inter-individual variability observed in the T1D population. It has been accepted by the FDA as a substitute to pre-clinical trials for closed-loop insulin delivery strategies and open-loop therapies [17].

The obtained dataset consists, for each individual, of one continuous time series, i.e., CGM; and two impulsive signals, i.e., INS and CHO, respectively. The first month of recordings was used to train and tune the model, while the second month was used for testing purposes. In particular, this choice allowed making a trade-off between model performance and the possibility of deploying such a model in real life. In fact, using more training data tends to improve model performance, but at the same time it requires more time to collect them, which might jeopardize patient’s adherence to the therapy.

### 2.4. Classification Results

Figure 3 shows the AUROC distributions computed in the test set for class C_1_, C_2_, C_3_, and their macro-average obtained in the population. On average, good performance is achieved. Specifically, the obtained average AUROC values are: AUROC(C_1_) = 0.89 [0.86, 0.93], AUROC(C_2_) = 0.76 [0.69, 0.82], AUROC(C_3_) = 0.86 [0.80, 0.91], AUROC(MACRO-AVERAGE) = 0.84 [0.78, 0.87]. Particularly, C_2_ results to be the most difficult class to predict. This is due to the discretization applied to G_min_. Specifically, since each G_min_ is hard-assigned to a single class, the classification error is likely higher when the original G_min_ was close to the th_hypo_/th_hyper_ thresholds before being discretized. Therefore, given that C_2_ is both upper (th_hyper_) and lower (th_hypo_) bounded, it is more likely that the error increases because of this discretization.

Notably, best performance is obtained for C_1_, meaning that the model is accurate in detecting hypoglycemia. Finally, several critical outliers are present in C_2_ and C_3_. However, they are strictly higher than 0.5, meaning that XGB always behaves better than the “random” classifier [24].

Finally, Figure 4 shows the receiving operator characteristic (ROC) curves obtained for a representative subject (adult#1). The model achieves very good performance. In particular, AUROC computed for class C_1_, C_2_, and C_3_ are 0.97, 0.83, and 0.95, respectively. Figure 4 reports the corresponding confusion matrix. Moreover, analyzing the classification error that XGB makes when it predicts the wrong class, it can be seen that it only picks adjacent classes. This is very important, since it avoids dangerous counter-actions by the patient, i.e., classifying an actual hypoglycemia as a hyperglycemia could lead the patient to increase the severity of the episode.

In the next section, an application of XGB is shown, the aim being to test its capability at improving glycemic outcomes, by using its outcome in real-time to adjust meal insulin boluses.

## 3. Application: Using the XGB Classifier to Adjust Meal Insulin Bolus

In a real-time setting, the developed XGB*_p_* classifier can be applied at meal-time to forecast the post-prandial glycemic status (i.e., hyperglycemia, euglycemia, or hypoglycemia). Such information on the future glycemic status can be used for multiple purposes. For instance, it can be used to generate smart alerts when future adverse events are forecasted; temporarily suspend basal insulin delivery, or suggest carbohydrate intake, to prevent hypoglycemia; and, the application which occupies us in this work, to recommend to modulate the insulin dose to be delivered at meal time. 

Here, we present a proof of concept for a simple empirical strategy to adjust the meal insulin bolus according to the real-time post-prandial glucose classification provided by the XGB model at meal-time. The aim of this study was to demonstrate the potential of XGB-based strategy at improving the standard T1D insulin therapy.

### 3.1. Meal Insulin Dose Adjustment Strategy

At meal time *t_m_*, if XGB classified future *G_min_* belonging to *C_i_*, IMB computed using (1) is adjusted using Equation (5), hereafter labeled as XGB-IMB:(5)$$IMB^{*}\left(t_{m}\right)\;=\;IMB\left(t_{m}\right)\;\times \;\left(1\;+\;p\left(C_{i}\right)\;\times \;f_{i}\left(IMB\left(t_{m}\right)\right)\right),$$
where *p*(*C*_*i*_) is obtained by XGB and denotes the probability of *G_min_* belonging to class *C_i_*; and *f_i_*(·) is an empirical modulation function associated to *C_i_*, which depends on the original IMB amount. In particular, *f_i_*(·) is defined in Equation (6).
(6)$$\begin{array}{l} f_{1}\left(IMB\left(t_{m}\right)\right)\;=\;\{\begin{array}{ll} f_{1_{LOW}} & if\;IMB\left(t_{m}\right)\;<\;5\;U\\ f_{1_{LOW}}\;+\;\frac{1}{3}\Delta f_{1} & if\;5\leq IMB\left(t_{m}\right)\;<\;10\;U\\ f_{1_{LOW}}\;+\;\frac{2}{3}\Delta f_{1} & if\;10\leq IMB\left(t_{m}\right)\;<\;15\;U\\ f_{1_{HIGH}} & if\;IMB\left(t_{m}\right)\;\geq \;15\;U \end{array}\\ f_{2}\left(IMB\left(t_{m}\right)\right)=\;0\\ f_{3}\left(IMB\left(t_{m}\right)\right)=\;\{\begin{array}{ll} f_{3_{LOW}} & if\;IMB\left(t_{m}\right)\;<\;5\;U\\ f_{3_{LOW}}\;+\;\frac{1}{3}\Delta f_{3} & if\;5\leq IMB\left(t_{m}\right)\;<\;10\;U\\ f_{3_{LOW}}\;+\;\frac{2}{3}\Delta f_{3} & if\;10\leq IMB\left(t_{m}\right)\;<\;15\;U\\ f_{3_{HIGH}} & if\;IMB\left(t_{m}\right)\;\geq \;15\;U \end{array} \end{array},$$
where $\Delta f_{i}\;=\;{\left(f_{i_{HIGH}}\;-\;f_{i_{LOW}}\right)}_{i\;=\;1,\;3}$. Intuitively, if C_2_ is predicted, no adjustment of IMB is performed. Moreover, by multiplying the modulation function by *p*(*C_i_*), the adjustment of IMB became directly proportional to how much XGB was “certain” about assigning the current input to class *C_i_*. This makes the adjustment robust to outliers, since in these cases, *p*(*C_i_*) results smaller, along with the IMB increment/decrement. Finally, the choice of using different adjustment depending on the original IMB amount allows the avoidance of too conservative, or too aggressive, adjustments when IMB is small, or big.

### 3.2. Simulated Scenario

XGB-IMB was tested in silico on a new dataset generated from the 100 virtual adult subjects of T1D-DM. Note that such a dataset is different from the one used to train and test the XGB model. In particular, two scenarios were created. In scenario A, 20 out of 100 subjects were used to tune the four XGB-IMB parameters ($f_{1_{LOW}}$, $f_{1_{HIGH}}$, $f_{3_{LOW}}$, and $f_{3_{HIGH}}$) over one-week simulation. Specifically, the parameter set was chosen by using a grid-search strategy, which uses a cost function that minimizes the average blood glucose risk index (BGRI) [30] of the studied population. In scenario B, the remaining 80 subjects were evaluated over one-month simulation with the aim of evaluating the performance of XGB-IMB strategy (with parameters obtained from scenario A). Finally, a comparison with the standard bolus calculator formula [31] was performed (SF-IMB).

### 3.3. Assessment of Glycemic Outcomes

For each subject of scenario B, performance of XGB-IMB and SF-IMB were compared in terms of: Mean (MEAN_BG_) and standard deviation of glucose concentration (SD_BG_); BGRI, percentage time in hypoglycemia (<70 mg/dL; %T_HYPO_); percentage time in hyperglycemia (>180 mg/dL; %T_HYPER_); percentage time in glucose target range [70, 180] mg/dL (%T_TARGET_); and percentage time in tight glucose target range [90, 140] mg/dL (%T_TTARGET_). These metrics are commonly used to assess glycemic outcomes in T1D and follow a recent consensus report on outcome measures for artificial pancreas clinical trials evaluation [32].

These population metrics are reported as mean (± standard deviation) for Gaussian distributed metrics or median [interquartile range]. For this purpose, non-Gaussianity of each distribution was checked by mean of the Lilliefors test with a 5% confidence level. Finally, assessment of statistical significance between-method differences was performed, with a 1% confidence level, using a paired *t*-test or the Wilcoxon rank sum test if the compared distributions were both Gaussian or not, respectively.

The four XGB-IMB parameters obtained from scenario A are *f_1LOW_* = −0.3, *f_1HIGH_* = −0.1, *f_3LOW_* = 0.8, and *f_3HIGH_* = 0.5. Note that, while XGB-IMB reduces IMB up to 30%/10% if future hypoglycemia (C_1_) is detected, it is quite aggressive at increasing IMB if future hyperglycemia is predicted (C_3_), i.e., up to 80%/50% if the original IMB is smaller/bigger than 5/15 units. 

Table 1 shows the population results obtained in the 80 subjects and corresponding to scenario B. Overall, XGB-IMB improves glycemic control across the population. In particular, when comparing SF-IMB vs. XGB-IMB, a statistically significant (*p* < 0.01) reduction of the overall glucose control in terms of MEAN_BG_ and SD_BG_ was achieved, i.e., 6.07 mg/dL and 4.95 mg/dL, respectively. Additionally, XGB-IMB lowered the glycemic risk in terms of BGRI (*p* > 0.01) by 1.36. Consistently, percentage time metrics showed better results when XGB-IMB was used. In detail, a significant improvement (*p* < 0.01) in %T_HYPER_ (5.34%, from 35.18% to 29.84%), %T_TARGET_ (5.02%, from 61.98% to 67.00%), and %T_TTARGET_ (2.95%, from 28.22% to 31.17%) was observed, while a non-significant improvement (*p* = 0.34) of %T_HYPO_ (0.11%, from 1.93% to 1.82%) was observed.

## 4. Discussion and Conclusions

In type 1 diabetes (T1D) management, continuous glucose monitors, insulin pumps, and activity monitors, together with additional physiological data streams (e.g., carbohydrate counting) can be used to develop new algorithms aimed at improving standard insulin therapy [33,34]. In this work, a new methodology based on a state-of-the-art machine learning model, i.e., the extreme gradient-boosted tree (XGB) model, is used to predict at meal-time postprandial glycemic status. Preliminary results obtained using in silico data generated with a state-of-the-art T1D simulator, show that XGB is accurate at discriminating between the selected post-prandial glycemic classes (i.e., hypoglycemia, hyperglycemia, and euglycemia). 

The XGB model outcome can potentially be used for different purposes (e.g., glucose alerts, decision support on insulin therapy). Here, we investigated how the prediction for the postprandial glycemic status at meal-time can be used to adjust the meal insulin bolus dose. Results obtained through a simple set of heuristic rules to adjust the meal insulin bolus, confirm that the proposed technique has the potential to improve post-prandial glycemic control in a T1D population. From a practical point of view, the proposed XGB algorithm could be easily integrated in currently available insulin pumps, or implemented in a stand-alone mobile application.

The presented study has some limitations that need to be addressed in future work. Machine learning techniques assume that data is independent and identically distributed. Thus, given the finite nature of the dataset used to train the model, wrong predictions may occur due to the approximations done at representing the underneath input–output correlation. To solve this issue, safety constraints need to be devised to improve the robustness of the XGB model and avoid misclassification errors. Another important aspect that needs to be further studied is the feature selection process. Given the inherent capability of XGB at automatically discriminating important features, no preliminary feature selection was performed during the data preparation phase. However, future work is needed to investigate if it is possible to reduce its dimensionality and improve the interpretability of the results. Furthermore, additional work can be done to develop a more specific and optimized policy to adjust the meal insulin bolus according to XGB classifications. Finally, although the obtained results using a state-of-the-art in silico environment are very encouraging, these are still preliminary results. Therefore, the next logical step will certainly involve the validation of the proposed technique using retrospective clinical data and its subsequent evaluation in a prospective clinical trial.

## Figures and Tables

**Figure 1 sensors-19-03168-f001:**
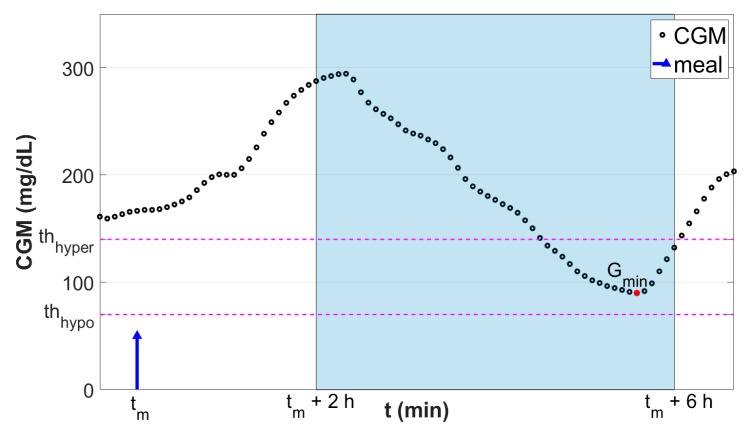
Graphical representation of the classification scheme. Black dots are the continuous glucose monitoring (CGM) samples. Blue stem indicates the meal event. Red dot is the minimum glucose level G_min_ reached in the fixed postprandial window (t_m_ + 2 h, t_m_ + 6 h; highlighted in light blue). Horizontal magenta lines denote the thresholds used to discretize G_min_ into the classification target *Y*.

**Figure 2 sensors-19-03168-f002:**
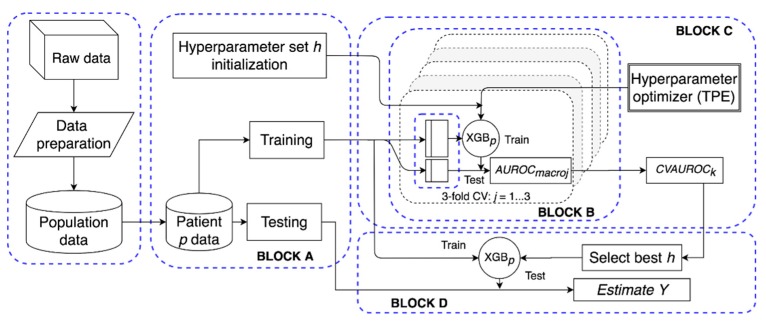
Structure of the proposed extreme gradient-boosted tree (XGB*_p_*) software framework. After data preparation, for patient *p*, block A initializes *h* and splits the data in training and test; block B computes the performance of the hyperparameter set in a three-fold cross validation (CV) setting over the training set; block C implements a tree-structured Parzen estimator to optimize *h*; block D selects the best *h* set and evaluates the performance of XGB*_p_* on the test set.

**Figure 3 sensors-19-03168-f003:**
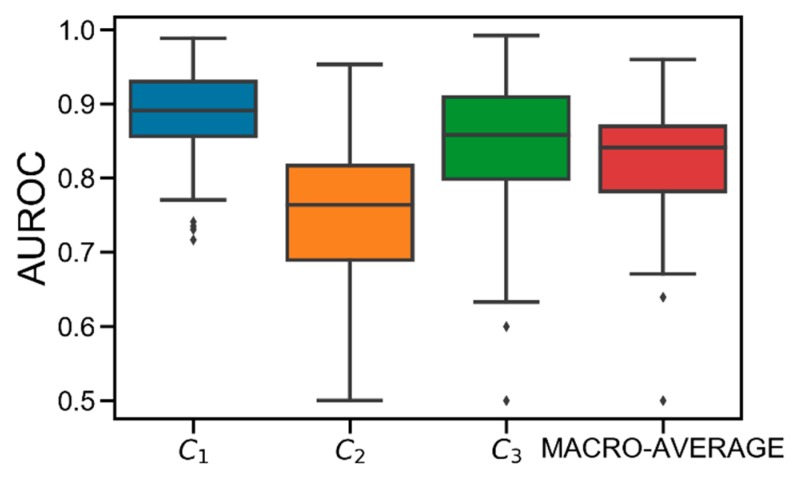
Boxplot representation of the distribution of area under the receiver operator characteristic curve (AUROC) obtained for C_1_ (in blue), C_2_ (in orange), C_3_ (in green), and their macro-average (in red) obtained in the population. Black horizontal line represents median, the black box marks the interquartile range, vertical black lines are the whiskers, and black diamonds indicate outliers.

**Figure 4 sensors-19-03168-f004:**
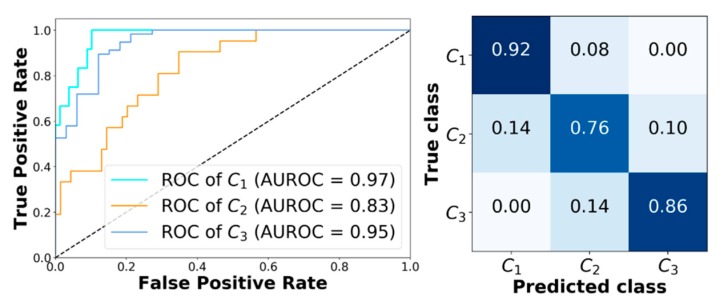
Classification results obtained with XGB corresponding to adult#1. Left panel: Receiving operator characteristic (ROC) curves obtained with XGB for C_1_, C_2_, and C_3_. Right panel: The corresponding confusion matrix.

**Table 1 sensors-19-03168-t001:** Obtained on 80 virtual adult subjects on Scenario B.

	SF-IMB	XGB-IMB	*P*-VALUE
**MEAN_BG_**	**167.12** [155.28, 181.16]	**161.05** [151.74, 169.33]	**<0.01 ***
**SD_BG_**	**54.97** [48.03, 63.95]	**51.02** [44.92, 59.97]	**<0.01 ***
**BGRI**	**9.36** [7.06, 11.43]	**8.00** [6.60, 9.83]	**<0.01 ***
**%T_HYPO_**	**1.93** [0.07, 3.81]	**1.82** [0.09, 3.81]	**0.34**
**%T_HYPER_**	**35.18** (±14.06)	**29.84** (±11.50)	**<0.01 ****
**%T_TARGET_**	**61.98** (±13.89)	**67.00** (±11.54)	**<0.01 ****
**%T_TTARGET_**	**28.22** [18.54, 40.56]	**31.17** [24.49, 42.60]	**<0.01 ***

**Results obtained using SF-IMB and XGB-IMB.** Median [interquartile range] is reported for MEAN_BG_, SD_BG_, BGRI, %T_HYPO_, %T_TTARGET_; mean (± standard deviation) are reported for %T_HYPER_, %T_TARGET_. *****: Statistically significant difference between XGB-IMB and SF-IMB using the Wilcoxon rank sum test. ******: Statistically significant difference between XGB-IMB and SF-IMB using the *t*-test.

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
