# Peer review of "Classification of Postprandial Glycemic Status with Application to Insulin Dosing in Type 1 Diabetes—An In Silico Proof-of-Concept"

_sensors, 2019, doi:10.3390/s19143168_

Reviewer 1 Report

This paper describes a methodology to classify glycemic status of type 1 diabetic patients and which is perhaps more interesting, an application to insulin dosing. the paper is very well written and organized and i really enjoy reading it. The XGB strategy is well explained along with the features an all the decisions are justified. the originality of the work is more in the combination of XBG qith an heuristic.

My only concern is related with the data used in the experiments. Although I know that is commonly used in the community, I dont see the benefit of using data from a simulator, instead of real retrospective data. Of course, data from a simulator could be more extensive, but since it is generated with a mathematical model, you can introduce some bias in the study. Apart from this, which is really a personal opinion, I will accept the paper in its current form

Author Response

This paper describes a methodology to classify glycemic status of type 1 diabetic patients and which is perhaps more interesting, an application to insulin dosing. the paper is very well written and organized and I really enjoy reading it. The XGB strategy is well explained along with the features an all the decisions are justified. The originality of the work is more in the combination of XBG with an heuristic.

My only concern is related with the data used in the experiments. Although I know that is commonly used in the community, I don’t see the benefit of using data from a simulator, instead of real retrospective data. Of course, data from a simulator could be more extensive, but since it is generated with a mathematical model, you can introduce some bias in the study. Apart from this, which is really a personal opinion, I will accept the paper in its current form.

We would like to thank Reviewer 1 for the very stimulating and positive feedback on our work. We are really pleased to hear that the Reviewer enjoyed reading our manuscript.

Reviewer 2 Report

The authors aim was to develop a classification technique used at meal time to predict future glycemic status with the eXtreme Gradient boosted tree (XGB).  They showed its usefulness and when tested in silico it performed better that a standard bolus calculator.  Figure 4 was impressive showing no hypoglycemia when none was predicted but the improvements using the XGB were perhaps less impressive in Table 1

The window used post meal is 2 to 6 hours.  The peak of glucose after meals is more like 70 min and the half life of the analogue short acting insulins is much shorter than the older human insulin so my question is - Are the authors missing  the main action time of the meal time insulin and really are they seeing the impact of the long acting insulin when looking at their time frame?

If one is going to study up to 6 hours post meal would an iterative feature using CGM eg glucose is dropping post prandially either food is taken or glucagon infused in a dual pump model of care

Activity can have a huge impact on glucose as mentioned on P2 L70 but from what I can make out is not used in the XGB, I know it may be difficult to quantify but is there a reason it is being left out?  If there is a good reason it should be stated in the discussion.

Author Response

The authors aim was to develop a classification technique used at meal time to predict future glycemic status with the eXtreme Gradient boosted tree (XGB).  They showed its usefulness and when tested in silico it performed better that a standard bolus calculator.  Figure 4 was impressive showing no hypoglycemia when none was predicted but the improvements using the XGB were perhaps less impressive in Table 1

We would like to thank Reviewer 2 for the insightful comments which have been of great help in improving the quality of the manuscript. Below we answer the questions the Reviewer raised. The corresponding changes in the manuscript have been highlighted in red.

Question 1: The window used post meal is 2 to 6 hours.  The peak of glucose after meals is more like 70 min and the half life of the analogue short acting insulins is much shorter than the older human insulin so my question is - Are the authors missing  the main action time of the meal time insulin and really are they seeing the impact of the long acting insulin when looking at their time frame?

We agree with the reviewer that the reason for choosing the postprandial time window of 2 to 6 hours post meal is not justified in the manuscript and we have addressed it as follows:

“The rationale for choosing a postprandial time window from two to six hours after meal time is that we are interested in computing the minimum postprandial glucose after glucose peak time. Note that the typical glucose peak after meals is around 70 minutes but can potentially be longer for low-absorption meals. We then chose to end the window at six hours post meal to make sure that the effect of the short-acting meal insulin bolus is over [19]. Finally, the assumption that basal insulin (i.e., long-acting insulin) is correctly adjusted is made. “

[19] Walsh, J., Roberts, R., & Heinemann, L. (2014). Confusion regarding duration of insulin action: a potential source for major insulin dose errors by bolus calculators. Journal of diabetes science and technology, 8(1), 170-178.

Question 2: Activity can have a huge impact on glucose as mentioned on P2 L70 but from what I can make out is not used in the XGB, I know it may be difficult to quantify but is there a reason it is being left out?  If there is a good reason it should be stated in the discussion.

Thank you for pointing this out. We agree with the Reviewer that this point is not clearly explained in the manuscript.  As suggested, we have clarified it in Section “2.3 Simulated dataset” by adding the following sentence:

“Of note, physical activity was not included in the simulations, since the current version of the UVa/Padova T1D Simulator does not include a mathematical model describing its impact on glucose metabolism.”